# Fractionalized conductivity and emergent self-duality near topological phase transitions

Yan-Cheng Wang [1], Meng Cheng[2], William Witczak-Krempa [3,4] & Zi Yang Meng [5✉]

The experimental discovery of the fractional Hall conductivity in two-dimensional electron gases revealed new types of quantum particles, called anyons, which are beyond bosons and fermions as they possess fractionalized exchange statistics. These anyons are usually studied deep inside an insulating topological phase. It is natural to ask whether such fractionalization can be detected more broadly, say near a phase transition from a conventional to a topological phase. To answer this question, we study a strongly correlated quantum phase transition between a topological state, called a $\mathbb{Z}_2$ quantum spin liquid, and a conventional superfluid using large-scale quantum Monte Carlo simulations. Our results show that the universal conductivity at the quantum critical point becomes a simple fraction of its value at the conventional insulator-to-superfluid transition. Moreover, a dynamically self-dual optical conductivity emerges at low temperatures above the transition point, indicating the presence of the elusive vison particles. Our study opens the door for the experimental detection of anyons in a broader regime, and has ramifications in the study of quantum materials, programmable quantum simulators, and ultra-cold atomic gases. In the latter case, we discuss the feasibility of measurements in optical lattices using current techniques.

[1] School of Materials Science and Physics, China University of Mining and Technology, Xuzhou, China. [2] Department of Physics, Yale University, New Haven, CT, USA. [3] Département de physique, Université de Montréal, Montréal, QC, Canada. [4] Centre de Recherches Mathématiques, Université de Montréal, Montréal, QC, Canada. [5] Department of Physics and HKU-UCAS Joint Institute of Theoretical and Computational Physics, The University of Hong Kong, Hong Kong SAR, China. ✉email: zymeng@hku.hk

Correlated topological phases exhibit phenomena that extend beyond the conventional paradigms of condensed matter physics, namely Landau's Fermi liquid theory for metals and the Landau–Ginzburg–Wilson symmetry-breaking scheme for phases and transitions. These topological phases are the embodiment of intrinsic topological order[1], and call for a deeper understanding of states of matter. Topologically ordered systems exhibit new types of particles called anyons that are neither fermions nor bosons. Some of these anyons can be used to robustly encode and manipulate quantum information, thus offering a viable platform for quantum computation[2].

Topological order was experimentally discovered in two-dimensional electron gases (2DEGs) under strong magnetic fields by measuring a well-known observable: the Hall conductivity. In the simplest integer quantum Hall state, the Hall conductivity is universally quantized as $\sigma_{xy} = e^2/h$, where $e$ is the electron charge and $h$ Planck's constant. In contrast, in fractional quantum Hall states[3,4], $\sigma_{xy}$ remains universal but becomes a fraction of $e^2/h$ as a consequence of the fractionalization of the electron. One prominent example is the fractional Hall state with $\sigma_{xy} = \frac{1}{3}\frac{e^2}{h}$ where the electrons fractionalize into anyons of charge $e/3$. Subsequent experiments, such as shot noise analysis of the edge modes[5,6], and thermal Hall conductance[7] have confirmed the fractionalized nature of the excitations. Unfortunately, many other equally interesting topologically ordered systems do neither possess a universal Hall response nor robust edge states. A representative example is the $\mathbb{Z}_2$ quantum spin-liquid (QSL), or in the bosonic language, a topologically ordered insulator[8], which could arise in frustrated magnets, bosonic Mott insulators or Rydberg atom-based programmable quantum simulators[9–15]. Unlike a regular paramagnet that would host bosonic spin-waves, this spin liquid has emergent bosonic and fermionic excitations, which carry 1/2 of the spin quanta of the spin waves, as well as visons, which are fluxes of an emergent gauge field. As the fluxes can only take two inequivalent values, the gauge field is said to be of $\mathbb{Z}_2$ type. A continuous transition between such a state and a conventional one is bound to be beyond the usual Landau–Ginzburg–Wilson paradigm. Direct experimental detection of fractionalization in these systems has remained an outstanding challenge[16–20].

Here, we propose a new experimental signature for fractionalization in $\mathbb{Z}_2$ QSLs that can be obtained already at the transition point from a conventional phase. As a concrete example, we consider a system in proximity to a quantum critical transition from a $\mathbb{Z}_2$ QSL to an ordinary superfluid, as shown in Fig. 1. Despite being gapless, the system's longitudinal conductivity becomes a simple fraction, 1/4, of its value at the usual quantum critical point between a trivial paramagnet and a superfluid. This fraction is a direct consequence of the fractionalization of the charge carriers at the quantum critical point (QCP), which carry 1/2 of the unit charge of the microscopic bosons. "Charge" here refers to the boson number (or spin, in a Mott insulator of electrons), so the bosons effectively split in two at the transition and in the quantum spin liquid, which is illustrated schematically in Fig. 1b. In addition, we uncover a crossover from a particle-like dynamical conductivity at low temperature, to a vortex-like one at higher temperatures. At an intermediate temperature denoted by $T_*$, the dynamical conductivity becomes nearly frequency-independent signaling the emergence of a self-dual quantum fluid. We argue that this striking behavior reveals the presence of visons, which are otherwise challenging to observe as they do not carry charge (spin). In the Discussion, we argue that these signatures can be observed using existing techniques in ultracold atomic gases loaded in an optical kagome lattice.

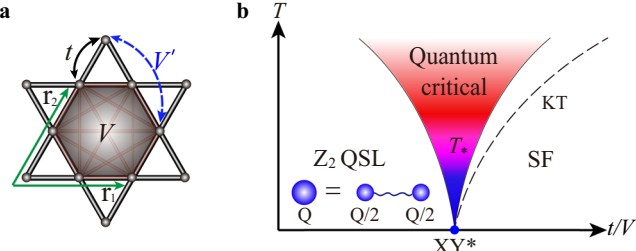

**Fig. 1 Kagome model for a topological phase transition. a** Kagome lattice with lattice vectors $\mathbf{r}_{1,2}$. The hopping $t$ and the interaction $V$ are given in the Hamiltonian Eq. (1). **b** Phase diagram of the kagome model as a function of $t/V$, and temprature $T$. The $\mathbb{Z}_2$ quantum spin liquid (QSL), superfluid (SF), and the XY* quantum critical point characterizing the ground state are shown. The charge fractionalization of spinons is schematically illustrated. In the quantum critical fan, blue indicates a particle-like conductivity, while an unconventional vortex-like response is in red. A nearly self-dual dynamical conductivity emerges at intermediate temperatures (purple) signaling the presence of visons. The same color scheme is used in representing the conductivity data in Figs. 2 and 3. The SF long-range order only exists at zero temperature; a Kosterlitz–Thouless (KT) transition, denoted by the black dashed line, separates the paramagnetic phase from the one with quasi-long-range order.

## Results

**Topological phase transition on the kagome lattice**. We consider the following Balents–Fisher–Girvin (BFG) model for bosons on a kagome lattice[9–14], depicted in Fig. 1a:

$$
\begin{aligned}
H = &-t \sum_{\langle ij \rangle} \left( b_i^\dagger b_j + \text{h.c.} \right) - \mu \sum_i n_i \\
&+ V \left( \sum_{\langle ij \rangle} n_i n_j + \sum_{\langle\langle ij \rangle\rangle} n_i n_j + \sum_{\langle\langle\langle ij \rangle\rangle\rangle} n_i n_j \right)
\end{aligned}
\tag{1}
$$

where $b_i^\dagger$ ($b_i$) creates (annihilates) a hard-core boson at site $i$, and $n_i = b_i^\dagger b_i$ measures the number of bosons therein. The $t$ term hops bosons between neighboring sites and the $V$ terms are repulsive interactions between any two bosons on a hexagon, see Fig. 1a. By the mapping $b_i^\dagger(b_i) \to S_i^+(S_i^-)$ and $n_i - 1/2 \to S_i^z$, the Hamiltonian can also be cast into an XXZ spin-1/2 model, with the chemical potential $\mu$ corresponding to external magnetic field $h$. We work primarily at the filling factor of $\langle n_i \rangle = 1/2$, i.e., 1/2 bosons at every site on average. The $\langle n_i \rangle = 1/3$ filling will be discussed below, and in that case, another repulsion $V'$ between the same sublattice sites on the neighboring hexagons is added to stabilize the QSL phase[13]. The Hamiltonian (1) conserves the total number of bosons, which corresponds to a U(1) symmetry. Accordingly, $b_i^\dagger$ creates an excitation of charge 1, which is the fundamental unit of charge in the system, in analogy with the charge of an electron in a solid.

As shown in Fig. 1b, at large $t/V$, the bosons can hop freely and will Bose–Einstein condense to form a superfluid at low temperature. In contrast, when the repulsion dominates an insulator will result, in which case the bosons become effectively frozen. Large-scale quantum Monte Carlo (QMC) simulations have shown that this quantum-phase transition occurs at $(t/V)_c = 0.070756(20)$[10,12]. So far, these properties seem conventional. However, the striking feature is that the insulator is a topological state of matter with fractionalized particles. Indeed, the emergent excitations do not carry charge 1 as expected, but rather 1/2: they are, heuristically speaking, half-bosons. The charge 1 bosons becomes fractionalized into pairs of bosons (called spinons) with half the fundamental charge. This is the

analog of emergent charge $e/3$ particles in a fractional quantum Hall state at filling 1/3. In fact, there are three types of topologically nontrivial emergent quasiparticles in the $\mathbb{Z}_2$ QSL: a bosonic spinon which carries half-integer charge, a bosonic vison with integer charge (including zero) but carrying $\pi$ flux of the emergent $\mathbb{Z}_2$ gauge field, and their bound state—a fermionic spinon.

Furthermore, the quantum-phase transition itself is highly unconventional. While it can be intuitively understood as the Bose–Einstein condensation transition of bosonic spinons, the symmetry-breaking paradigm of Landau–Ginzburg cannot explain the emergent fractionalization. Interestingly, the transition is continuous meaning that quantum critical fluctuations proliferate to large scales, and can thus amplify signatures of fractionalization.

Using large-scale quantum Monte Carlo (QMC) simulations we search for such signatures using a key observable, the conductivity. This is in part motivated by the fundamental role that conductivity has played in the discovery of fractional quantum Hall states. Since time-reversal is not broken here, the Hall conductivity vanishes and we are left with the longitudinal conductivity, denoted by $\sigma$. One couples the system to an external potential that causes a flow of charge (bosons), and the conductivity is given by the linear response expression $\sigma(\omega) = -\frac{i}{\omega}\langle J_x(\omega)J_x(-\omega)\rangle$, where we have allowed for a drive oscillating with frequency $\omega$. In the Discussion, we explain how this is possible using current techniques in ultracold atomic gases. $J_x(\omega)$ is the usual boson current along the $x$ direction (denoted by the lattice vector $\mathbf{r}_1$ in Fig. 1a) at frequency $\omega$. In the QMC simulations, one has directly access to imaginary frequencies $\omega \to i\omega_n = i2\pi Tn$, where $n$ is an integer and $T$ the temperature. An important challenge is the continuation from imaginary to real frequencies. Reliable numerical techniques for this purpose, such as stochastic analytic continuation[21,22] which we will use in this work, are under active development and have been successfully employed in various quantum many-body systems[14,23–25].

In Fig. 2, we show the conductivity of the system at the quantum critical coupling $(t/V)_c = 0.070756(20)$ and filling $\langle n_i \rangle = 1/2$. We compute $\sigma(\omega_n)$ with system sizes $L = 12, 24, 36, 48, 60, 72, 96$ and inverse temperature $\beta V = 300, 350, 390, 400, 450, 500, 600$ (with statistical errors obtained from QMC simulations and standard data fitting; this also applies to the data shown in Fig. 3). QMC simulations and conductivity measurements are described in "Methods", and additional details, especially the two-step extrapolation of $\sigma(L \to \infty, \beta \to \infty)$ to the thermodynamic limit, are given in Supplementary Note 2. We plot the finite-temperature conductivity and extrapolate it to $L \to \infty$ and then to $\beta \to \infty$ ($T \to 0$), as shown by the black solid dots, and $\sigma$ is then expected to become a universal scaling function $f(\omega/T)$, or in imaginary frequencies, $f(i\omega_n/T)$[26,27]. In the low-temperature regime $\omega_n \gg T$, the conductivity should saturate to its ground-state constant value $\sigma(\infty)$. This plateau is clearly observed in Fig. 2, and the resulting conductivity obeys a striking relation:

$$\sigma_{\mathrm{XY}^*}(\infty) = \frac{1}{4}\,\sigma_{\mathrm{XY}}(\infty), \qquad (2)$$

where XY denotes the conventional superfluid-to-insulator transition which is of the XY universality, and since the transition in our model involves fractionalization, it is denoted as XY*[9,28,29]. The XY transition arises in non-frustrated lattices, the canonical example being the Bose–Hubbard model on the square lattice at unit filling, which has been experimentally realized with ultracold atoms[30]. Comparing our numerical value $\sigma_{\mathrm{XY}^*}(\infty) = 0.098(9)$ with the best estimate for that of the XY transition $\sigma_{\mathrm{XY}} = 0.3554$[31–36] gives a ratio

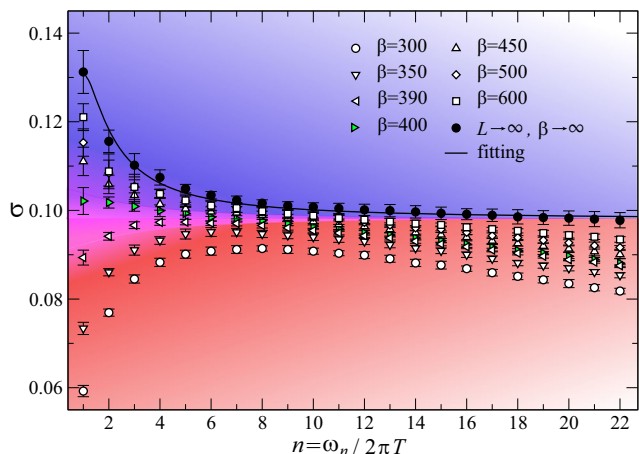

**Fig. 2 Conductivity fractionalization and dynamical self-duality at half-filling.** Longitudinal conductivity data, including a two-step extrapolation from finite sizes and temperatures to the low-$T$ thermodynamic limit, of our kagome model at boson filling $\langle n_i \rangle = 1/2$ as a function of $n = \omega_n/2\pi T$ at the XY* critical point $(t/V)_c = 0.070756(20)$ with $L = 12, 24, 36, 48, 60, 72, 96$ and $\beta V = 300, 350, 390, 400, 450, 500, 600$. In the thermodynamic limit, $L \to \infty$ and then $\beta \to \infty$, the universal constant $\sigma_{\mathrm{XY}^*}(\infty)$ appears, and we obtain a plateau value of $\sigma_{\mathrm{XY}^*}(\infty) = 0.098(9)$ which is $\frac{1}{4}\sigma_{\mathrm{XY}}(\infty)$. At finite temperatures, we see a crossover from a particle-like conductivity (blue) to a vortex-like one (red). In the vicinity of $\beta_* \approx 400$ (c.f. the green triangles), we observe a nearly constant dynamical conductivity indicating a self-dual-quantum fluid with visons.

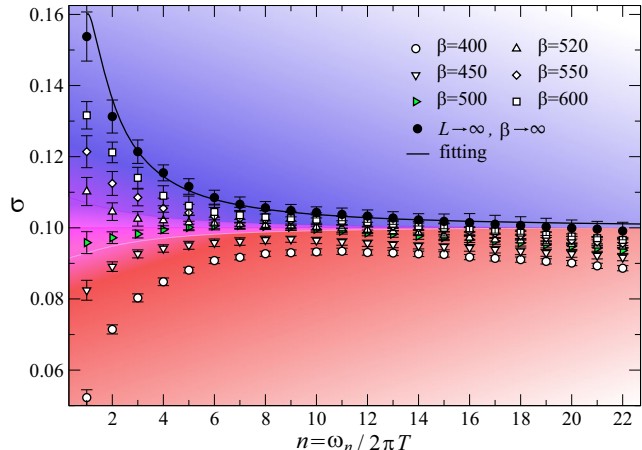

**Fig. 3 Topological signatures at 1/3 filling.** The conductivity data of our kagome lattice model at boson filling $\langle n_i \rangle = 1/3$ as a function of $n = \omega_n/2\pi T$ at the XY* critical point $(t/V)_c = 0.07773(5)$, with $V'/V = 0.005$, $L = 12, 24, 36, 48, 60, 72$ and $\beta V = 400, 450, 500, 520, 550, 600$. In the thermodynamic limit $L \to \infty$ and then $\beta \to \infty$, the universal constant $\sigma_{\mathrm{XY}^*}(\infty)$ appears, and we obtain $\sigma_{\mathrm{XY}^*}(\infty) = \frac{1}{4}\sigma_{\mathrm{XY}}(\infty)$ as discussed in the text. As in Fig. 2, we see a thermal crossover from a particle-like conductivity (blue) to a vortex-like one (red). The self-dual regime occurs around $\beta_* \approx 500$ (c.f. the green triangles).

$\sigma_{\mathrm{XY}^*}(\infty)/\sigma_{\mathrm{XY}}(\infty) = 0.27(3)$, which is 1/4 within error bars. The suppression of the conductivity at the fractionalized XY* critical point compared to its XY counterpart is given by a simple rational number, 1/4, which is reminiscent of the fractional Hall conductivity observed in 2DEGs—also a rational fraction of the conductivity at unit filling.

We now turn to the case of $\langle n_i \rangle = 1/3$ filling. It was shown in ref. [13] that a XY* QCP also occurs between $\mathbb{Z}_2$ QSL and

superfluid phases, when the aforementioned $V'$ term is added to the Hamiltonian to stabilize the QSL. The $\mathbb{Z}_2$ QSL in this case has identical topological order as the one at 1/2 filling, and spinons still carry half U(1) charge[12,14,20]. Although the emergent $\mathbb{Z}_2$ gauge field sees a different background charge density for 1/2 and 1/3 fillings, we expect this subtle difference do not affect the critical properties at the $XY^*$ transition. The QMC results are shown in Fig. 3, with system sizes $L = 12, 24, 36, 48, 60, 72$, and $\beta V = 400, 450, 500, 520, 550, 600$ at the critical point $(t/V)_c = 0.07773(5)$. The plateau in the conductivity, after the two-step extrapolation to the thermodynamic limit (as denoted by the black solid dots), also yields $\sigma_{XY^*}(\infty) = 0.100(13)$ and $\sigma_{XY^*}(\infty)/\sigma_{XY} = 0.28(4)$, which is again $\frac{1}{4}$ within error bars.

As we shall now explain, the fractionalized conductivity observed here and the fractional Hall conductivity observed in 2DEGs share a common origin: charge fractionalization.

**From fractionalized charge to fractional conductivity**. To understand the aforementioned results at the $XY^*$ QCP, we can resort to a coarse-grained description in terms of a quantum field theory (see Supplementary Note 1 for a detailed review of this theory). A complex field $\phi$ is introduced to represent the emergent bosonic spinons. Since a conventional charge 1 boson is associated with a pair of spinons, we assign a unit charge to $\phi^2$. As such, the spinon field must carry charge $Q = 1/2$ under the U(1) particle conservation symmetry. The form of the Hamiltonian is then constrained by the fact that the critical theory has an emergent Lorentz invariance and takes the same form as for the regular XY transition: $H = \int d^2\mathbf{x}(|\partial_0\phi|^2 + |\nabla\phi|^2 + r|\phi|^2 + \lambda|\phi|^4)$, where $r$ tunes the system to criticality. It is important to note that physical observables must be composed of an even number of spinons. For instance, the superfluid corresponds to a Bose–Einstein condensate of conventional bosons, namely $\phi^2$. Since we are interested in conductivity, we need to first specify the form of the physical current:

$$\mathbf{J} = \frac{1}{2}i(\phi\nabla\phi^* - \phi^*\nabla\phi) \qquad (3)$$

which is 1/2 of the usual current one would get at the XY transition. The 1/2 ensures that the field describing the original bosons has a unit U(1) charge. It then follows from the linear response expression $\sigma = -\frac{i}{\omega}\langle J_x(\omega)J_x(-\omega)\rangle$ that the conductivity at the $XY^*$ transition is 1/4 that of its XY value, in perfect agreement with our numerical results, i.e., both at the $XY^*$ QCPs of $\langle n_i \rangle = 1/2$ in Fig. 2 and $\langle n_i \rangle = 1/3$ in Fig. 3. We note that the above argument is nonperturbative in the interaction strength since the $\mathbb{Z}_2$ gauge field, and the associated gapped visons, become nondynamical at asymptotically low temperatures.

**Visons and dynamical self-duality**. Besides probing the ground state at the quantum-phase transition, our results for the conductivity shown in Figs. 2 and 3 extend well into the quantum critical fan of Fig. 1b at finite temperature. This experimentally accessible regime offers an opportunity to probe strongly interacting quantum fluids in thermal equilibrium. Due to the emergent scale invariance at quantum criticality, the rate for excitations to relax is given by the absolute temperature $k_B T/\hbar$[37], where we have temporarily reinstated Boltzmann's and Planck's constants. As such, the finite frequency conductivity will be a scaling function of the frequency divided by this universal rate, $\sigma(\omega, T) = f(\omega/T)$, which holds at sufficiently low $T$ but for fixed $\omega/T$. We have obtained this universal scaling function at imaginary frequencies $f(i\omega_n/T)$, see the fit of the thermodynamic values in Figs. 2 and 3 (the fitting procedure is described in Supplementary Note 2). At large values of the argument, $f(i\omega_n/T)$

reduces to the ground-state conductivity, which is 1/4 the value of the ordinary XY QCP.

At smaller frequencies, the scaling function shows the same upturn previously obtained using QMC simulations for the regular XY QCP[31–35]. A response with such an upturn is referred to as particle-like[20,38,39] since it shares the same form as that of regular bosons in the XY universality class. In Figs. 2 and 3, we observe that as the quantum system is heated up, there is a gradual reduction of the upturn of the low-frequency conductivity. At sufficiently high temperatures, the conductivity acquires a downturn near the DC limit. We say that such a conductivity is the "dual" of the particle-like conductivity since under the usual particle–vortex duality, the dual vortices have a conductivity given by the inverse of that of the original bosons, $1/\sigma(\omega)$[40]. The duality thus converts an upturn into a downturn, yielding a vortex-like conductivity[38,39]. At an intermediate temperature that we call $T_*$, the dynamical conductivity becomes nearly frequency-independent as is shown in purple in Figs. 2 and 3. We refer to this type of conductivity as dynamically self-dual owing to the fact that under usual particle–vortex duality, a frequency-independent response remains flat[41]. This crossover from particle-like to vortex-like response within the quantum critical fan results from the geometrical frustration of the kagome Hamiltonian, and is absent at the conventional XY transition[31]. It is thus a striking new feature of the topological-phase transition.

In order to understand the origin of this striking phenomenon, we need to go back to the full cast of topological particles. Beyond the spinons described by the $XY^*$ field theory discussed above, the QSL also hosts charge-neutral visons[20]. As these are gapped, they decouple at asymptotically low temperatures. However, as we heat the system, the temperature approaches the gap scale of the visons, which we shall independently quantify below. The visons then become thermally excited and begin to interact and scatter the charge-carrying spinons. This new scattering channel leads to the observed reduction of the conductivity at low frequencies. As the visons are $\pi$-fluxes of the emergent gauge field, the spinons effectively move in a random background emergent magnetic field, which results in lower mobility. Remarkably, the dynamical conductivity in the self-dual regime is $\sigma(\omega) \approx \frac{1}{4}\sigma_{XY}(\infty)$ for all frequencies, extending the topological fractionalization to the dynamical regime. It is important to emphasize that we have written the conductivity in real frequencies since the analytic continuation can be trivially performed for a constant function, which is an advantage of the self-dual regime compared to temperatures away from $T_*$.

To further test the above conclusion regarding the vison signatures in the conductivity, we analyze the QMC dynamical density–density correlation function $\langle n_i(\tau)n_j(0)\rangle$ (or $\langle S_i^z(\tau)S_j^z(0)\rangle$ in the spin language), and stochastically analytically continue to real frequencies[14,22–24]. We expect the number density correlations to reveal properties about vison excitations, as deep inside the spin-liquid phase ($t/V \ll 1$) it can be shown that the number operator $n_i$ (or $S_i^z$) creates a pair of visons[9,14]. The vison gap should stay finite within the entire spin-liquid phase, including the QCP, so it is reasonable to expect that $n_i$ creates vison pairs near and at the transition. Figure 4a shows the spectrum for filling 1/2, while Fig. 4b is for filling 1/3. A fundamental feature of the spectra is the absence of excitations at low energies, which leads to the conclusion that the visons are gapped[14]. We have also verified the vison gap by directly measuring the exponential decay of the $\langle S_i^z(\tau)S_j^z(0)\rangle$ correlation in imaginary-time QMC data, and the obtained gaps are consistent with those read from the spectra in Fig. 4 (examples of the comparison are given in Supplementary Note 3). The gap in the spectrum gives twice the vison gap $\Delta_v$, since visons are always created in pairs. We thus estimate

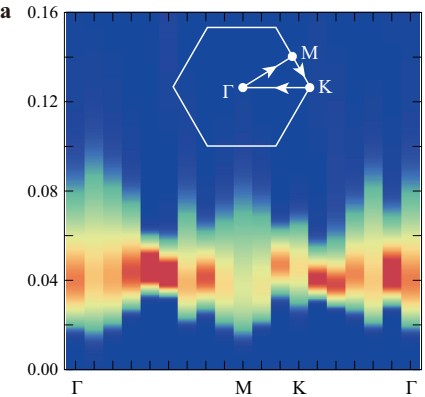
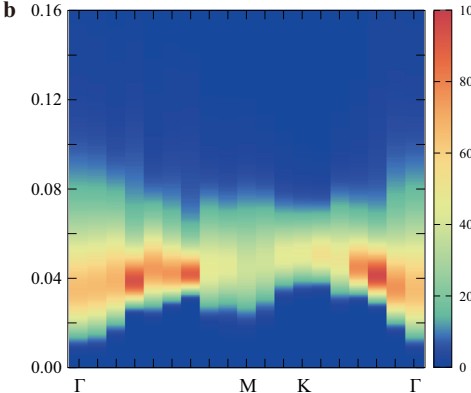

**Fig. 4 Probing the vison gap.** Vison-pair spectra obtained from the QMC dynamical density–density $\langle n_i(\tau)n_j(0)\rangle$ (or $\langle S_i^z(\tau)S_j^z(0)\rangle$ in the spin language) correlation function and stochastic analytic continuation, plotted along the high symmetry path $\Gamma$--$M$--$K$--$\Gamma$ in the Brillouin zone. The horizontal axis is momentum, while the vertical one represents energy (frequency) (note the energy scale here is very low compared with, for example, the scale of the bare interaction which is $V=1$). **a** At the XY* critical point $(t/V)_c = 0.070756$ at boson filling $\langle n_i\rangle =1/2$ with system size $L=18$ and $\beta=500$. **b** At the XY* critical point $(t/V)_c = 0.07773$ at filling $\langle n_i\rangle =1/3$ with system size $L=18$ and $\beta=600$. The vison-pair continua are clearly visible above (twice) the vison gap.

$\Delta_v \sim 0.01$ at filling 1/2, and $\sim 0.005$ at filling 1/3. We expect that $\Delta_v$ sets the scale for the self-dual temperature $T_*$ obtained for the conductivity. We indeed find that these two quantities are of the same order of magnitude, with $T_*$ being roughly a third of the vison gap. Furthermore, $\Delta_v$ is lower at 1/3 filling compared to half-filling, consistent with the fact that $T_*$ is smaller at 1/3 filling. It would be interesting to perform a more detailed theoretical analysis that would relate $T_*$ to the vison gap. This would require studying a field theory beyond the one for the pure XY* quantum critical point, since finite mass visons would need to be included at finite temperature.

## Discussion

We obtained the finite frequency conductivity near the unconventional XY* quantum critical point, which is associated with fractionalization, topological order, and an emergent $\mathbb{Z}_2$ gauge field. The topological-phase transition separates a $\mathbb{Z}_2$ QSL haboring fractionalized spinon and vison excitations from a conventional superfluid phase. We have shown that the ground-state conductivity reveals the existence of fractionalized charge, i.e., $\sigma_{XY^*}(\infty) = \frac{1}{4}\sigma_{XY}(\infty)$. This sharp signature in the conductivity is to be contrasted with other types of "indirect" measurements on QSLs such as inelastic neutron scattering that can only observe the spinon-pair continua, which is easily confused with the continua generated from disorder[42,43]. We have uncovered another qualitatively new signature, namely the crossover from a particle-like (DC upturn) to a vortex-like (DC downturn) dynamical conductivity as the system is heated up. Strikingly, at intermediate temperatures, we discovered a dynamically self-dual conductivity that is nearly independent of the frequency $\sigma(\omega) \approx \frac{1}{4}\sigma_{XY}(\infty)$. This is in sharp contrast to the usual XY transition, and results from the presence of thermally excited topological particles, the visons. Therefore, the conductivity fractionalization and emergent self-duality discovered in this work open the door for the experimental detection of fractionalized particles such as anyons in a variety of quantum materials, and ultracold atomic gases for example the recently proposed Rydberg atom-based programmable quantum simulators on the kagome lattice[15]. Recent experiments[44] in ultracold atomic gases have yielded the frequency-dependent conductivity for atoms loaded in a two-dimensional optical lattice, which is precisely what is needed to measure the conductivity predicted in our work. In the experiment, the alternating current is obtained by applying a

spatially uniform but temporally oscillating force via the displacement of a harmonic trapping potential. Since we predict the emergence of a dynamically self-dual regime, this should be easier to observe since it will be apparent in a wide range of frequencies and does not require that the system be cooled to the absolute lowest temperatures. We reiterate that the self-dual response holds at real frequencies, which is what is measured. We also note that bosonic atoms have been successfully loaded in an optical kagome lattice[45], so that all the basic experimental ingredients are present. It would be desirable to further modify the Hamiltonian in order to increase the vison gap, thus increasing $T_*$, and making the crossover more readily observable. Finally, it will be of interest to extend our findings to other QSL phases, as well as to certain non-Fermi liquids and their unconventional transitions. As a concrete example, it would be of interest to study the finite-temperature dynamical conductivity near the topological QCP that is "dual" to the one studied in this work: visons condense, while the spinons maintain a small gap throughout. An inversion of the observed crossover would be expected (vortex/particle-like at small/large $T$), but a detailed study is needed owing to the strongly interacting nature of the transition.

## Methods

We simulate the Hamiltonian in Eq. (1) on the kagome lattice by using a worm-type continuous-time QMC technique[46,47]. In the simulations, we take system sizes $L = 12, 24, 36, 48, 60, 72, 96$, and the inverse temperature $\beta V = 300$, 350, 400, 450, 500, 520, 550, 600. The conductivity $\sigma$ can be expressed as $\sigma(i\omega_n) = -\frac{i}{\omega_n}\langle J_x(\omega_n)J_x(-\omega_n)\rangle$ with $J_x(\omega_n)$ the current operator along the $x$ direction ($\mathbf{r}_1$ in Fig. 1a) of the kagome lattice. In the QMC simulations, the imaginary frequency conductivity $\sigma(i\omega_n)$ is computed as

$$\sigma(i\omega_n) = \frac{\langle -k_x\rangle - \Lambda_{xx}(i\omega_n)}{\omega_n}$$
$$= \frac{\langle |\sum_k P_k^x e^{i\omega_n \tau_k}|^2\rangle}{\beta L^2 \omega_n} \tag{4}$$

where $\langle k_x\rangle$ is the kinetic energy associated with the $x$-oriented bond, and $\Lambda_{xx}(i\omega_n)$ is the Fourier transform of imaginary-time current-current correlation function[48], and $\sum_k$ runs through the volume of $L\times L\times\beta$ of the QMC configurational space with $P_k^x$ denoting the projection of the $k$th hopping along the $x$ direction. A similar measurement of conductivity has been performed at the XY QCP[33].

In order to obtain real-frequency spectral functions, the stochastic analytic continuation (SAC) scheme is employed to obtain the spectral function $A(\mathbf{q}, \omega)$ from the imaginary-time correlation function $S(\mathbf{q}, \tau)$, $S(\mathbf{q}, \tau) = \frac{1}{\pi}\int_0^\infty d\omega A(\mathbf{q}, \omega)$ $(e^{-\tau\omega} + e^{-(\beta-\tau)\omega})$. It is known that the problem of inverting the Laplace transform is equivalent to find the most probable spectra $A(\omega)$ out of its exponentially many suggestions to match the QMC correlation function $S(\tau)$ with respect to its stochastic errors, and such transformation has been converted to a Monte Carlo sampling process[21,22]. This QMC-SAC approach has been successfully applied to

quantum magnets ranging from the simple square lattice Heisenberg antiferromagnet[23,25] to deconfined quantum critical point and quantum spin liquids with their fractionalized excitations[14,24].

## Data availability

The data that support the findings of this study are available from the corresponding author upon reasonable request.

## Code availability

All numerical codes in this paper are available upon reasonable request to the authors.

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

## Acknowledgements

We thank Kun Chen, Éric Dupuis, Snir Gazit, Yang Qi, Zhijin Li, David Poland, Subir Sachdev, and Erik Sørensen for insightful discussions. Y.C.W. acknowledges the supports from the NSFC under Grant No. 11804383, the NSF of Jiangsu Province under Grant No. BK20180637, and the Fundamental Research Funds for the Central Universities under Grant No. 2018QNA39. M.C. acknowledges support from NSF under award number DMR-1846109 and the Alfred P. Sloan Foundation. W.W.-K. was funded by a Discovery Grant from NSERC, a Canada Research Chair, a grant from the Fondation Courtois, and a "Établissement de nouveaux chercheurs et de nouvelles chercheuses universitaires" grant from FRQNT. ZYM acknowledges the RGC of Hong Kong SAR of China (Grant Nos. 17303019, 17301420, and AoE/P-701/20)), MOST through the National Key Research and Development Program (Grant No. 2016YFA0300502) and the Strategic Priority Research Program of the Chinese Academy of Sciences (Grant No. XDB33000000). We thank the Computational Initiative at the Faculty of Science and the Information Technology Services at the University of Hong Kong and the Tianhe supercomputing platforms at the National Supercomputer Centers in Tianjin and Guangzhou for their technical support and a generous allocation of CPU time. The authors acknowledge Beijng PARATERA Tech CO.,Ltd. (https://paratera.com/) for providing HPC resources that have contributed to the research results reported within this paper.

## Author contributions

M.C., W.W.-K., and Z.Y.M. initiated the work. Y.-C.W. performed the QMC calculations, Y.-C.W. and Z.Y.M. carried out the numerical data analysis. M.C. and W.W.-K. performed the theory analysis. All authors wrote the manuscript together.

## Competing interests

The authors declare no competing interests.
