## [Peer Review File · Nature Communications]

REVIEWER COMMENTS

Reviewer #1 (Remarks to the Author):

The authors calculate the longitudinal conductivity in a model Z₂ spin liquid using QMC simulations and show that it acquires a fractionalized value of the usual XY transition at very low temperatures, reflecting the possible fractionalized nature of excitations in the system.

Although the numerical simulations on the model Hamiltonian are interesting, there are a number of issues and questions which come up

1) What is the relevant temperature scale T^* in supposed model realizations in ultracold atoms? To claim experimental relevance, numbers on reasonable current setups would be useful. I suspect this temperature is incredibly low and hence it this would not be a realistic signature (there are numerous other proposed experimental signatures of fractionalization which also suffer from similar defects).

2) I'm not sure I understand all the claims about the duality (despite the references). The spinons are gapless (or have a much smaller gap than the visons) and the conductivity with only spinons at low temperatures will be different to a system with both spinons and visons at high temperatures. And where is the cross-over to a trivial paramagnetic regime?

3) I'm not convinced Fig. 4 shows a vison-pair continuum. The features are relatively flat. How much broadening is occurring due to temperature? And how much is an artifact of the method?

Reviewer #2 (Remarks to the Author):

This is an interesting paper discussing the interplay of topological quasiparticles in the conductivity of a Z_2 quantum spin liquid near a continuous transition driven by the condensation of one of the two species of quasiparticles present. Using a state of the art numerical approach (QMC + SAC) combined with field theoretic arguments, the authors are able to show that the conductivity "fractionalises" (namely reflecting the fact that it is carried by fractional quasiparticles), and it changes character from spinon-like to vison-like as a function of temperature in the critical fan. These theoretical results are interesting and important in their own right and may have direct implications to the study and detection of quantum spin liquid materials. I am therefore inclined to recommend the paper for publication in Nature Communications, provided that the authors address the following concern. The crossover from particle-like to vortex-like behaviour is very interesting but also somewhat puzzling to me, and I would like the authors to expand their discussion in the manuscript. To the best of my understanding, it should be possible to attribute particle like behaviour to the spinons and vortex-like to visons. The former are gapless at the QCP and it is sensible that they dominate the conductivity at low temperatures (and low ω ?); as temperature is increased above the vison gap, I can understand why the two contributions become comparable and may lead to a self-dual signal. But why does the conductivity become vortex dominated at high temperatures? Is it perhaps that the contribution from visons is stronger than from spinons, and therefore the self-dual point is not when they are equally probable to occur in the system? It may be that I am misunderstanding the particle and vortex-like contributions altogether, but it would be anyway good for the reader if the authors could clarify the matter in their paper.

As a side note, a similar transition driven by the condensation of visons instead of spinons should perhaps show a reverse particle-vortex behaviour?

Typos and minor points:

-- In Fig.2 the symbols should be made a little more visible, if possible. The authors refer to "black solid dots" which are in fact not black solid (this applies also to Fig.3). And also the self-dual line $\beta^*=400$ is difficult to spot -- could it be highlighted better?

-- the \cdot in Eq.(3) after $1/2$ should perhaps be removed (even though I think I understand the intent of the authors). Also, a reference to the relevant part of the Supplementary where the factor of $1/2$ is discussed further would be useful here.

-- typo: "fonction" on p.5

-- typo: "the obtained the gaps" on p.6

-- the acronym BFG should be introduced for the benefit of the reader

-- in the Suppl., "integrated charge \tilde{J}_0 inside" is misleading since " \tilde{J}_0 " is a charge density

-- in the Suppl., typo: "relative(ly) high temperature"

-- in the Suppl., I wonder if the fitting parameter ω should be replaced with a different symbol to avoid confusing the reader with angular frequency

-- in the Suppl., typos: "taking care (of) the corrections" and "actually post(pose) a challenging problem"

-- in the Suppl. p.4, the figure label should be S3... whereas Fig.4(b) is in the main text (this may need to be tidied up across the Suppl.)

-- in the Suppl., how were the fits performed in Fig.S3? How was the τ fitting integral chosen? Or are the straight lines drawn for specific gap values for comparison? The authors should clarify their discussion of the procedure

Response to Referee A

We would like to thank referee A for his/her high assessments of the importance of our work. Regarding the comments, we give our detailed responses below. We have also made the corresponding improvements in the revised manuscript.

Comment 1: - What is the relevant temperature scale T_* in supposed model realizations in ultracold atoms? To claim experimental relevance, numbers on reasonable current setups would be useful. I suspect this temperature is incredibly low and hence it this would not be a realistic signature (there are numerous other proposed experimental signatures of fractionalization which also suffer from similar defects).

Reply 1: We thank the referee for the careful reading of our manuscript. This question regarding the experimental temperature scales is important. Actually, the temperature or energy scales where topological phenomena emerge may not be that low. Near the QCP, the ratio of the boson hopping to the repulsion is $t/V \approx 0.07$. The self-dual temperature at half-filling is then $T_* \sim V/400 \sim t/28$, so around 0.03 of the boson kinetic energy. Moreover, one of our main discoveries is that the vortex-like behavior with a suppression at low frequencies occurs at temperatures *above* T_* , as can be seen in Fig.2. This suppression is contrary to what one expects for the regular XY superfluid-insulator QCP. As such, the relevant temperature for the emergence of topological phenomena is above 5% of the boson hopping.

In addition, even if such temperature/energy scale might be low for the present setting, this is also because the scale is determined by the size of the vison gap. Our work serves as a proof of principle that has succeeded in finding fractionalized conductivity and emergent self-duality near topological phase transitions in a non-trivial yet simple lattice model. One could modify the model in order to increase the vison gap, which would in turn lead to a bigger T_* .

Comment 2: - I'm not sure I understand all the claims about the duality (despite the references). The spinons are gapless (or have a much smaller gap than the visons) and the conductivity with only spinons at low temperatures will be different to a system with both spinons and visons at high temperatures. And where is the cross-over to a trivial paramagnetic regime?

Reply 2: We thank the referee for these insightful questions. First, as the temperature is increased, more visons become excited, and they interact with and scatter the spinons. This leads to a suppression of the conductivity at smaller frequencies, and the observed dynamically self-dual conductivity (i.e. independent of the frequency). It is important to emphasize that the spinons and visons are strongly coupled, and obtaining a more detailed understanding of the crossover will require substantial theoretical advances. Second, the cross-over scale to the fully trivial paramagnetic regime will occur at a much higher temperature scale $T \sim V$, and is not shown in the paper. We have focused on the low temperature regime, where novel topological phenomena are observed.

Comment 3: - I'm not convinced Fig. 4 shows a vison-pair continuum. The features are relatively flat. How much broadening is occurring due to temperature? And how much is an artifact of the method?

Reply 3: We again thank the referee for this insightful comment. The flatness of the vison-pair continuum in the Balents-Fisher-Girvin kagome model is known from previous studies as well, for example, Refs. [10,14] in the revised manuscript. Since the temperature is low, $T = V/500$ for 1/2-filling and $T = V/600$ for 1/3-filling, the energy scale of the spectra are already very small, i.e., the vison-pair gap at 0.02 in Fig.4, which is consistent with the fitting of the vison-pair gap from

the raw data of the imaginary-time correlation functions in QMC, as denoted in the Sec.3 of the Supplementary Material. Also, please note that the broadness might also appear amplified because we are showing a narrow energy range.

As for the QMC-SAC method for extracting the real-frequency spectra, which is developed over the past decades, it has been verified in many works on quantum magnetic systems and have been directly compared with the Bethe ansatz, exact diagonalization, field theoretical analysis and even experiments, such as the works on 1D Heisenberg chain [Phys. Rev. E 94, 063308 (2016)], 2D Heisenberg model [Phys. Rev. X 7, 041072 (2017)], Z_2 quantum spin liquid model with fractionalized spectra [Phys. Rev. Lett. 121, 077201 (2018), npj Quantum Materials 6, 39 (2021)] and quantum Ising model with direct comparison with neutron scattering and NMR experiments [Nature Communications 11,5631 (2020) and Nature Communications 11, 1111 (2020)]. It has been proved to provide the reliable spectral information for magnon, spinon, vison and other dynamical correlation functions.

Response to Referee B

We thank referee for his/her support for the publication of our work.

Comment 1: - This is an interesting paper discussing the interplay of topological quasiparticles in the conductivity of a Z_2 quantum spin liquid near a continuous transition driven by the condensation of one of the two species of quasiparticles present. Using a state of the art numerical approach (QMC + SAC) combined with field theoretic arguments, the authors are able to show that the conductivity "fractionalises" (namely reflecting the fact that it is carried by fractional quasiparticles), and it changes character from spinon-like to vison-like as a function of temperature in the critical fan. These theoretical results are interesting and important in their own right and may have direct implications to the study and detection of quantum spin liquid materials. I am therefore inclined to recommend the paper for publication in Nature Communications, provided that the authors address the following concern.

Reply 1: We thank the referee for the positive and concise summary of our work.

Comment 2: - The crossover from particle-like to vortex-like behaviour is very interesting but also somewhat puzzling to me, and I would like the authors to expand their discussion in the manuscript. To the best of my understanding, it should be possible to attribute particle like behaviour to the spinons and vortex-like to visons. The former are gapless at the QCP and it is sensible that they dominate the conductivity at low temperatures (and low ω ?); as temperature is increased above the vison gap, I can understand why the two contributions become comparable and may lead to a self-dual signal. But why does the conductivity become vortex dominated at high temperatures? Is it perhaps that the contribution from visons is stronger than from spinons, and therefore the self-dual point is not when they are equally probable to occur in the system? It may be that I am misunderstanding the particle and vortex-like contributions altogether, but it would be anyway good for the reader if the authors could clarify the matter in their paper.

Reply 2: This is a central point of the work, and it is important to clarify it further, as the referee recommends. First, the conductivity is strictly speaking always due to the spinons, since they carry charge while the visons do not. The essential point is that as T is increased, more visons become excited, and they scatter the spinons. This leads to a suppression of the conductivity at smaller frequencies, and the observed crossover. In the main paper, we explain that the visons act like a

random dynamical magnetic field for the spinons. We have further added statements clarifying the charge-neutrality of visons, and the effect of the random magnetic field on the charge mobility.

Regarding the self-dual regime near T_* , it happens when there are enough visons to flatten the original spinon conductivity. As it is a strongly interacting QCP, it is difficult to anticipate *exactly* at what temperature this will happen. Our observations show that T_* is close to the vison-gap scale, but the proportionality constant will require a much more advanced treatment that is beyond the scope of the current work.

Comment 3: - As a side note, a similar transition driven by the condensation of visons instead of spinons should perhaps show a reverse particle-vortex behaviour?

Reply 3: The referee brings up a very interesting point here. If the visons condense at the QCP instead of the spinons, the analysis will be quite different since the visons don't carry charge. The charge response should show a strong suppression at low frequencies due to the spinon gap. We can thus say that the low-T conductivity would be vortex like, as the referee points out. As the system heats up above the spinon-gap scale, a particle-like response (DC upturn) would be expected due to the excited spinons. Interestingly, since the visons do interact strongly with the spinons, the crossing of the QCP will have non-trivial consequences on the conductivity. It would be most interesting to study this in a concrete model. We have added two sentences to present the idea at the end of the discussion. We thank the referee for this interesting suggestion.

Comment 4: - Typos and minor points: ...

Reply 4: We thank the referee for catching these typos. We have made the changes accordingly.

List of changes in the revised manuscript

Below we list the changes in the revised manuscript. The main changes to the text are marked in red in the manuscript.

1. In response to the comment of Referee B, we have revised and elaborated the discussions about the emergent dynamic duality in the revised manuscript.
2. Building on the question of Referee B, we have added a discussion at the end of the paper about a different QCP, where visons condense instead of the spinons.
3. Typos, pointed out by the referees, have been adjusted.
4. Some references have been updated or added.

REVIEWERS' COMMENTS

Reviewer #1 (Remarks to the Author):

The authors have not really answered my questions by talking about how the work is a proof of principle (whereas my question was whether the proof of principle is relevant at any reasonable experimental scale currently accessible).

The same applies to the second question where they have said future theoretical work is needed to understand the actual conductivity at T^* and above (which they claim is more experimentally relevant from their reply to comment 1).

I am not convinced about the experimental relevance of the work as of yet, but the work does contain some interesting numerical results. I will leave the decision up to the editor.

Reviewer #2 (Remarks to the Author):

The authors have addressed the points that I had raised in my referee report and I am satisfied with the answers and changes in the manuscript. I am happy to recommend it for publication in your journal.

Response to Referee A

We would like to thank referee A for his/her high assessments of the importance of our work. Regarding the comments, we give our detailed responses below. We have also made the corresponding improvements in the revised manuscript.

Comment 1: The authors have not really answered my questions by talking about how the work is a proof of principle (whereas my question was whether the proof of principle is relevant at any reasonable experimental scale currently accessible).

The same applies to the second question where they have said future theoretical work is needed to understand the actual conductivity at T^* and above (which they claim is more experimentally relevant from their reply to comment 1).

I am not convinced about the experimental relevance of the work as of yet, but the work does contain some interesting numerical results. I will leave the decision up to the editor.

Reply 1: We thank the referee for his/her careful and critical reading of our manuscript. In our first reply, we argued that the qualitatively new signatures will emerge at temperatures *above* 5% of the boson hopping amplitude, t . Looking at the trends in our data, this could readily reach 10%, and potentially more. These are low temperatures for cold atomic gases loaded in optical lattices, but they are not entirely unreasonable. We agree with the referee that it would be important to improve the model in order to increase the vison gap scale, hence increase the temperatures at which the novel signatures appear. We added a comment about this point in the Discussion.

With regards to the explanation of the conductivity at and above T_* , we provide sound arguments for the scale T_* , and the suppression of the conductivity above it in the manuscript. The conceptual work required to quantitatively explain the finite-temperature dynamical conductivity in a strongly interacting quantum critical system is substantial. In fact, no controlled framework exists at present. We believe that our numerical findings, together with the key conceptual elements we have identified, will help move this frontier forward. However, this lies beyond the scope of our paper, which already contains substantial numerical and conceptual analysis.

List of changes in the revised manuscript

Below we list the changes in the revised manuscript.

1. We modified the format of the manuscript and added Data availability Section, Code availability Section, Author contributions Section, and Competing interests Section to meet Nature Communications requirements.
2. We added a sentence in the Discussion: “It would be desirable to further modify the...”
3. We added the definition of error bars of the data in Fig. 2, Fig.3, and Supplementary Figs. 1-3.